# Improved TV Image Denoising over Inverse Gradient

**Minmin Li, Guangcheng Cai \*, Shaojiu Bi and Xi Zhang**

Faculty of Science, Kunming University of Science and Technology, Kunming 650500, China
* Correspondence: guangchengcai@sina.com

**Abstract:** Noise in an image can affect one's extraction of image information, therefore, image denoising is an important image pre-processing process. Many of the existing models have a large number of estimated parameters, which increases the time complexity of the model solution and the achieved denoising effect is less than ideal. As a result, in this paper, an improved image-denoising algorithm is proposed based on the TV model, which effectively solves the above problems. The $L_1$ regularization term can make the solution generated by the model sparser, thus facilitating the recovery of high-quality images. Reducing the number of estimated parameters, while using the inverse gradient to estimate the regularization parameters, enables the parameters to achieve global adaption and improves the denoising effect of the model in combination with the TV regularization term. The split Bregman iteration method is used to decouple the model into several related subproblems, and the solutions of the coordinated subproblems are derived as optimal solutions. It is also shown that the solution of the model converges to a Karush–Kuhn–Tucker point. Experimental results show that the algorithm in this paper is more effective in both preserving image texture structure and suppressing image noise.

**Keywords:** image denoising; inverse gradient; regularization parameter; split Bregman iterative method; Karush–Kuhn–Tucker condition

## 1. Introduction

Symmetry in a mathematical functional equation refers to a transformation or operation, including the operators in fractional calculus, fractal, and local fractional calculus that leave the equation unchanged [1]. In image processing, the transformation and inverse transformation of matrices, mapping and inverse mapping, Fourier transform, and inverse variation all fall under the category of symmetry. The concept of symmetry is therefore widely used in image denoising.

Image denoising is an active research problem in the field of image processing. Common noises generated during image acquisition or transmission include Gaussian noise, impulse noise, Poisson noise, etc. These noises often lead to image quality degradation. The process of recovering a clean, high-quality image from the observed low-quality noisy image $f : \Omega \to R$ is known as image denoising, and this process is reversible. Here, the image domain $\Omega$ denotes the bounded connected open set $R^2$, which has a Lipschitz boundary. The process of directly recovering a noisy image is undesirable because it lacks some a priori information about the image. Over the last few decades, researchers have proposed many solutions to this problem and have applied these methods to problems such as noise removal from acquired images in seismic surveys [2], hyper parameterized model identification [3], and fault estimation [4]. Among these, regularization methods are widely used in image processing [5,6] and have produced a general form of Gaussian noise removal models.

$$\min_u \left\{ \frac{\lambda}{2} \|u - f\|_2^2 + R(u) \right\} \tag{1}$$

here, $R(u)$ denotes the regularization term, which is typically used to describe a priori information about the image, such as smoothness, continuity, and bounded variation.

$\lambda$ denotes the regularization parameter, which is used to balance $\|u - f\|_2^2$ and $R(u)$. The data fidelity term $\|u - f\|_2^2$ has a penalty on the reconstructed image $u$. The aim is to bring the observed noisy image $f$ closer to the original high-quality image by solving model (1).

In model (1), the key to improving the denoising ability of the model is the correct choice of the regularization term $R(u)$. The intrinsic features of an image can be obtained by using model-based methods [7,8], discriminative-based methods [9], and variational-based methods [10,11] to find suitable a priori information about the image. This paper focuses on variational-based approaches. The classical variational model is the ROF model the fully variational TV model proposed by Rudin et al. The TV model can be used for image restoration [12], image reconstruction [13,14], blind deconvolution [15], and vector-valued images [16]. Although the TV model provides better results, it still does not describe the local details of the image very well. The specific reason is that, in numerical experiments, the difference grid of the TV model depends only on the horizontal and vertical directions and does not guarantee the Euler–Lagrange equations of the response spread in the direction of the edge tangents. In addition, the TV model does not preserve the geometric features of the image and is prone to step artifact effects in flat regions. To overcome these drawbacks, many improvement models have been proposed. For example, Lui et al.'s deep learning-based image segmentation method effectively solved the problem of adhesion and overlap between adjacent particles of mineral images [17]. Zhou et al. proposed a radical unsupervised remote sensing image classification method to further improve the classification accuracy of the model [18]. Zhao et al. designed a residual network structure by dividing the input feature map into two parts, which reduced the network parameters and improved the network inference speed [19]. In another example, Wang et al. proposed a fractional-order TV model, which has served better in color image denoising and decomposition [20]. Kazemi Golbaghi et al. proposed a new fractional-order full variational model, a model whose order depends on the image automatically assigned to it and is better able to capture the edges and details of the image [21]. Lian et al. proposed a non-convex fractional-order television model that has an excellent ability to overcome step effects and maintain neat contours [22]. Although these methods can eliminate step artifacts to some extent, they are computationally complex and often result in phenomena such as blurring of edges or leaving residual noise. To further improve the performance of the model, a well-known approach is to use second-order (or higher-order) variations instead of TV terms, which also avoids step artifact effects to some extent. Duan et al. decomposed the image into two parts, structure and texture, and proposed an edge-weighted second-order variational model for image decomposition; this model has improved recovery over the TV model [23]. Fang et al. combined convolutional neural networks with traditional variational models, using relevant edge features obtained from noisy images as a priori information to make the models strongly adaptive [24]. Phan et al. used bounded Hessian regularize to eliminate step effects and preserve image edge structure [25]. However, the second-order variational model, while avoiding the step artifact effect to some extent, may not be as effective as the first-order variational model in terms of denoising. In addition, the number of parameters [26] involved in estimating the model is larger, and solving all of them at the same time is difficult. This fact motivates us to find methods that reduce the number of parameters but still have a better denoising effect. In other words, the proposed model not only removes step artifact effects better and retains more image information, but also simplifies the computational complexity of the parameters.

The main contributions of this paper are: firstly, the inclusion of the $L_1$ parameter regularization term makes the solution of the TV model more sparse, and, to some extent, also improves the step artifact effect generated by the TV model. Secondly, the number of parameters to be estimated is reduced and the inverse gradient multiscale is used to capture the edges of the image to estimate the regularization parameters, making the parameters globally adaptive. Applying this regularization parameter to the model together with the TV regularization term can significantly improve the quality of image denoising. Finally, the split Bregman iteration method (SBIM) is used to decouple the multivariate

image denoising model into several related solvable subproblems, and the solutions of the coordinated subproblems are obtained as optimal solutions to the original problem. It is also shown that the solution of the model converges to a Karush–Kuhn–Tucker (KKT) point.

## 2. Related Work

The proposed algorithm is closely related to the regularization term, so this section focuses on several image-denoising problems that contain different regularization terms.

### 2.1. The ROF Model

The classical ROF model, proposed by Rudin et al. in 1992, can be represented in the form of a minimized energy generalization function as follows.

$$\underset{u}{\arg\min}\left\{\frac{\lambda}{2}\|u - f\|_2^2 + \|\nabla u\|_1\right\} \tag{2}$$

Here, $\|\nabla u\|_1$ denotes the TV regularization term. The TV term is capable of suppressing solutions where the model produces oscillations and discontinuities and is often used to deal effectively with image edges. However, the degree of diffusion of model (2) in the local normal direction is always zero, which also usually leads to the generation of segmental constant solutions. In other words, the TV regularization term tends to give rise to step artifact effects in the model.

### 2.2. p-Order TV-Based Model

In order to overcome the step artifact effect produced by the ROF model, some researchers have introduced a p-order full variational-based model, which can be expressed as follows.

$$\underset{u}{\arg\min}\left\{\frac{\lambda}{2}\|u - f\|_2^2 + \|\nabla^p u\|_1\right\} \tag{3}$$

When $p = 2$ and $\|\nabla^2 u\|_1 = \sum_{i=1}^{m}\sum_{j=1}^{l}\sqrt{u_{xx}(i,j)^2 + u_{xy}(i,j)^2 + u_{yx}(i,j)^2 + u_{yy}(i,j)^2}$, model (3) is a higher order full variance model, about which there are applications such as the Laplacian penalty [27,28], the anisotropic second-order regularization [29], the Hessian Schatten-norm regularization [30], etc. In fact, the segmented linear solution generated by the second-order derivative of the segmented vanishing can be a better fit for smooth intensity changes. As a result, such models have better denoising performance than TV models in terms of maintaining smooth regions. However, this model tends to lead to blurred edges. When $0 < p < 1$, model (3) is transformed into a fractional-order variational model, which is also better at removing step artifact effects, but has a higher computational complexity.

### 2.3. Lasso Regression Model

The Lasso regression model was first proposed by Robert Tibshirani. The model allows for variable selection and complexity adjustment (regularization) when fitting a generalized linear model. Thus, the Lasso regression model can be used to solve approximate solutions to the original problem regardless of whether the target-dependent variable is continuous, binary, or multivariate discrete. In image processing, the Lasso regularization term allows the model to produce a sparse matrix of weights for feature selection. To a certain extent, it can also prevent the underfitting of the model. The model can be expressed as follows.

$$\underset{u}{\arg\min}\|f - Au\|_2^2 + \lambda\|u\|_1 \tag{4}$$

However, the model is not derivable everywhere and it is not possible to obtain a solution to the model by direct derivation. Some researchers have used the coordinate descent method and the Least Angle Regression method for solving.

## 3. The Proposed New Model

### 3.1. New Models

Typically, images have different structures in different regions. Image denoising aims to remove noise while retaining as much structural information as possible. Whereas the TV regularization term can effectively preserve the structural information of the image, the lasso regularization term can improve the sparsity of the model solution. Therefore, we propose an improved TV model.

$$\underset{u}{\arg\min} \left\{ \frac{1}{2} \|Au - f\|_2^2 + \alpha \|\nabla u\|_1 + \beta \|u\|_1 \right\} \tag{5}$$

Model (5) is effective in removing image noise, preserving image edge information, and eliminating step artifacts produced by the conventional TV model and its variants. However, parameter estimation for this model is a major challenge and the effectiveness of the step artifact removal depends on the robust parameters chosen. In model (5) we consider a simultaneous multiplication by $\kappa$, $\kappa > 0$, and set $\beta = \kappa \alpha$, which gives the following form.

$$\underset{u}{\arg\min} \left\{ \frac{\kappa}{2} \|Au - f\|_2^2 + \beta \|\nabla u\|_1 + \beta \kappa \|u\|_1 \right\} \tag{6}$$

$$\underset{u}{\arg\min} \left\{ \frac{\kappa}{2\beta} \|Au - f\|_2^2 + \|\nabla u\|_1 + \kappa \|u\|_1 \right\} \tag{7}$$

To simplify the parameters, let $\lambda = \frac{\kappa}{\beta}$, then the following model can be obtained.

$$\underset{u}{\arg\min} \left\{ \frac{\lambda}{2} \|Au - f\|_2^2 + \|\nabla u\|_1 + \kappa \|u\|_1 \right\} \tag{8}$$

In model (8), $\lambda$ is the parameter for the data fidelity term and $\kappa > 0$ is the equilibrium regularization parameter. The model has the following advantages: (i) in practice, the equilibrium parameter $\kappa$ of model (8) plays a preferential role in eliminating noise or facilitating the elimination of step artifacts; (ii) the parameters $\lambda$ of the model are easier to estimate than $\alpha$ and $\beta$ of model (5).

In solving the model, estimating the values of the parameters is an important task. In [31,32] it was demonstrated that the simultaneous use of the inverse gradient-driven parameters with the TV regularization term can significantly improve the denoising quality of damaged images. Therefore, this paper further improves the denoising performance of the model by introducing a multi-scale inverse gradient adaptive regularization parameter that depends on the noisy image, which is expressed as follows.

$$\lambda(f) = \frac{\mu}{1 + \tau \max_\rho |G_\rho * \nabla f|_2^2} \tag{9}$$

Here, $G_\rho = \frac{1}{2\pi\rho^2} \exp(-\frac{x_1^2 + x_2^2}{2\rho^2})$ is a two-dimensional Gaussian kernel function, * denotes a two-dimensional convolution operation, $\tau$ is a constant taking value in the range $10^{-4} \sim 10^{-2}$ and $\mu = 2/9$. For the scale parameter $\rho$, only five scale levels are considered in this paper, $\rho = 1, 2, 3, 4, 5$, respectively. The regularization parameter $\lambda(f)$ is globally adaptive to noisy images. Thus, model (8) can be rewritten as.

$$\underset{u}{\arg\min} \left\{ \frac{\lambda(f)}{2} \|Au - f\|_2^2 + \|\nabla u\|_1 + \kappa \|u\|_1 \right\}, \tag{10}$$

the value of the scaling parameter $\rho$ corresponds to the value of $|G_\rho * \nabla f|_2$. Of all the values $|G_\rho * \nabla f|_2$, we choose the maximum value as the value of the parameter $\lambda(f)$ estimated by the model.

### 3.2. Solving the Model

It can be seen that model (10) is a larger-scale non-convex optimization problem and solving model (10) directly is difficult. An effective numerical solution method is SBIM, which is commonly used to solve high-dimensional signal processing problems such as machine learning, computer vision, image, and signal processing. This method is closely related to the dual decomposition, the alternating direction multiplier method, and the Dykstra's alternating projection method, etc. SBIM involves decomposing a larger global problem into a series of solvable local sub-problems related to variables and computing the global solution by using the solutions of the sub-problems.

In this paper, model (11) is solved by iteratively updating the original variables and the corresponding dual variables of the augmented Lagrange function. Specifically, this paper introduces auxiliary variables $v$ and $w$ then reformulates model (10) as the following constrained optimization problem.

$$\left\{ \begin{array}{c} \underset{u}{\operatorname{argmin}} \left\{ \frac{\lambda(f)}{2} \|Au - f\|_2^2 + \|\nabla u\|_1 + \kappa \|u\|_1 \right\} \\ s.t.\ v = \nabla u\,,\ w = u \end{array} \right\} \tag{11}$$

To simplify the process of solving model (11), this paper introduces two dual variables (Lagrange multipliers) $y = (y_1, y_2)^T$ and $p = (p_1, p_2)^T$. The problem is then reformulated as.

$$\begin{aligned} &\underset{u,v,w,y,p}{\operatorname{minmax}} L(u, v, w, y, p) \\ &= \frac{\lambda(f)}{2} \|Au - f\|_2^2 + \|v\|_1 + \langle y, v - \nabla u \rangle + \frac{\tau_1}{2} \|v - \nabla u\|_2^2 + \kappa \|w\|_1 + \langle p, w - u \rangle + \frac{\tau_2}{2} \|w - u\|_2^2\,, \end{aligned} \tag{12}$$

where $L(u, v, w, y, p)$ denotes the augmented Lagrange function $\tau_1$ and $\tau_2$ denote the penalty parameters. Further rewriting of the problem (12) yields.

$$\begin{aligned} &\underset{u,v,w,b_1,b_2}{\operatorname{minmax}} L(u, v, w, b_1, b_2) \\ &= \frac{\lambda(f)}{2} \|Au - f\|_2^2 + \|v\|_1 + \frac{\tau_1}{2} \left\| v - \nabla u - b_1^k \right\|_2^2 + \kappa \|w\|_1 + \frac{\tau_2}{2} \left\| w - u - b_2^k \right\|_2^2, \end{aligned} \tag{13}$$

here, $b_1 = -y/\tau_1$ and $b_2 = -p/\tau_2$. All the variables in problem (13) are difficult to solve simultaneously because all of them are coupled together. If SBIM is used multiple variables of the problem can be decoupled into corresponding sub-problems. At this point, one can consider fixing the other variables and solving the subproblems for each variable to obtain the optimal solution to the original problem, as shown in Algorithm 1.

---

**Algorithm 1: SBI to solve the problem (13).**

**Input:**

(1) Set parameters $\kappa$, $\tau_1$ and $\tau_2$;
(2) initialization: original values of $u^0$, $v^0$, $w^0$, $b_1^0$, $b_2^0$;
(3) Iterate (14a)–(14e) below until stopping criterion is met;

$$\left\{ \begin{array}{ll} u^{k+1} := \operatorname{argmin}_u L\left(u, v^k, w^k, b_1^k, b_2^k\right) & \text{(14a)} \\ v^{k+1} := \operatorname{argmin}_v L\left(u^k, v, w^k, b_1^k, b_2^k\right) & \text{(14b)} \\ w^{k+1} := \operatorname{argmin}_w L\left(u^k, v^k, w, b_1^k, b_2^k\right) & \text{(14c)} \\ b_1^{k+1} := \operatorname{argmin}_{b_1} L\left(u^k, v^k, w^k, b_1^k, b_2^k\right) & \text{(14d)} \\ b_2^{k+1} := \operatorname{argmin}_{b_2} L\left(u^k, v^k, w^k, b_1^k, b_2^k\right) & \text{(14e)} \end{array} \right.$$

**Output:** $u := u^{k+1}$ as the restored image.

---

The computational efficiency of Algorithm 1 depends on how well the individual subproblems are solved with high accuracy. The subproblems represented by (14a)–(14e) are shown to be solved as follows.

3.2.1. Solution of Related Sub-Problems

(1)   Subproblem (14a). This subproblem is a smooth convex optimization problem and can be expressed as.

$$u^{k+1} = \text{argmin}_u \left\{ \frac{\lambda(f)}{2} \left\| Au^k - f \right\|_2^2 + \frac{\tau_1}{2} \left\| v^k - \nabla u^k - b_1^k \right\|_2^2 + \frac{\tau_2}{2} \left\| w^k - u^k - b_2^k \right\|_2^2 \right\} \tag{15}$$

The Euler–Lagrange equations for this problem can be obtained by using the variational method.

$$(\lambda(f)A^T A + \tau_1 \nabla^T \nabla + \tau_2 \cdot I) u^{k+1} = \lambda(f)A^T f + \tau_1 \nabla^T v^k - \tau_1 \nabla^T b_1^k + \tau_2 w^k - \tau_2 b_2^k \tag{16}$$

For different boundary conditions, the solution process of the linear Equation (16) corresponds to different numerical methods. The Laplace operator $\Delta$ is semi-negative definite when using the zero Neumann boundary condition or the zero Dirichlet boundary condition. In this case, the preprocessed conjugate gradient (PCG) method is the solution that can be used. In this paper, the boundary conditions used are assumed to be periodic, then problem (16) can be solved using fast Fourier variation.

$$u^{k+1} = F^{-1} \left( \frac{F(\lambda(f)A^T f + \tau_1 \nabla^T v^k - \tau_1 \nabla^T b_1^k + \tau_2 w^k - \tau_2 b_2^k)}{F(\lambda(f)A^T A + \tau_1 \nabla^T \nabla + \tau_2 \cdot I)} \right) \tag{17}$$

Here, $F(\cdot)$ and $F^{-1}(\cdot)$ denote the fast Fourier variation and its inverse variation.

(2)   Subproblem (14b). This subproblem can be expressed as.

$$v^{k+1} = \text{argmin}_v \left\{ \left\| v^k \right\|_1 + \frac{\tau_1}{2} \left\| v^k - \nabla u^k - b_1^k \right\|_2^2 \right\} \tag{18}$$

It can be found that problem (18) is a convex optimization problem and, according to Theorem 1, the local solution of this problem can be solved by a threshold operator.

$$v^{k+1} = shrink(\nabla u^k + b_1^k, \frac{1}{\tau_1}) = \frac{\nabla u^k + b_1^k}{\left| \nabla u^k + b_1^k \right|} \max(\left| \nabla u^k + b_1^k \right| - \frac{1}{\tau_1}, 0) \tag{19}$$

**Theorem 1.** *For a convex optimization problem*

$$X^{k+1} = \text{argmin}_v \left\{ \alpha \|X\|_1 + \frac{\beta}{2} \|X - Y\|_2^2 \right\}$$

*the solution to this problem is defined* $[0, 1]^2$ *and can be formulated concretely as.*

$$X^{k+1} = shrink(Y, \frac{\alpha}{\beta}) = \frac{Y}{|Y|} \cdot \max(|Y| - \frac{\alpha}{\beta}, 0)$$

(3)   Subproblem (14c). Subproblem (14c) can be expressed as.

$$w^{k+1} = \text{argmin}_w \left\{ \kappa \left\| w^k \right\|_1 + \frac{\tau_2}{2} \left\| w^k - u^k - b_2^k \right\|_2^2 \right\} \tag{20}$$

Similarly to subproblem (14b), this subproblem is also a convex optimization problem and one of its local solutions can be expressed in the following form.

$$w^{k+1} = shrink(u^k + b_2^k, \frac{\kappa}{\tau_2}) = \frac{u^k + b_2^k}{\left| u^k + b_2^k \right|} \max(\left| u^k + b_2^k \right| - \frac{\kappa}{\tau_2}, 0) \tag{21}$$

3.2.2. Update of the Multiplier

It can be noted that the multipliers $b_1$ and $b_2$ are functions on the dual variables $y$ and $p$, respectively, and the process of iteratively updating the multipliers is also the process of iteratively updating $y$ and $p$. The use of SBIM to solve model (10) is accompanied by the generation of multipliers. The multipliers should be updated as the subproblem is continuously updated. A total of 2 multipliers are involved in this paper and the corresponding iterative update equations are given below.

$$\begin{cases} b_1^{k+1} = 2b_1^k + \nabla u^k - v^k \\ b_2^{k+1} = 2b_2^k + u^k - w^k \end{cases} \tag{22}$$

## 4. Convergence Analysis

In this section, we perform a convergence analysis of the proposed algorithm based on Theorem 2 [33].

**Theorem 2.** *A basic model for non-negative matrix decomposition is to use the least squares loss function to measure the approximation of the matrix, resulting in the following standard non-negative matrix decomposition problem.*

$$\min f(X, Y) \triangleq \frac{1}{2}\|XY - M\|_2^2 \; s.t. \; X \geq 0, Y \geq 0 \tag{23}$$

Let $\left\{Z^k\right\}_{k=1}^{\infty}$ be the sequence generated by using SBIM on (23). If $\left\{Z^k\right\}_{k=1}^{\infty}$ satisfies the condition $\lim\limits_{k \to \infty} (z^{k+1} - z^k) = 0$, then the cumulative point $\left\{Z^k\right\}_{k=1}^{\infty}$ is a KKT point of model (23).

Assume that $U$ and $V$ are auxiliary variables introduced in the process of solving model (23) and that $\Lambda$ and $\Pi$ are Lagrange multipliers. Define a relevant six-tuple $Z \triangleq (X, Y, U, V, \Lambda, \Pi)$, then the KKT condition that model (23) should satisfy is shown below.

$$\begin{cases} (XY - M)Y^T + \Lambda = 0 \\ X^T(XY - M) + \Pi = 0 \\ \quad X - U = 0 \\ \quad Y - V = 0 \\ \Lambda \leq 0 \leq U, \Lambda \odot U = 0 \\ \Pi \leq 0 \leq V, \Pi \odot V = 0 \end{cases} \tag{24}$$

Based on the splitting operator, the Lagrange function for the problem (12) can be expressed as.

$$\begin{aligned} &L(u^{k+1}, v^{k+1}, w^{k+1}, y^{k+1}, p^{k+1}) \\ &= \frac{\lambda(f)}{2}\|Au - f\|_2^2 + \|v\|_1 - \langle y^T, v - \nabla u\rangle + \frac{\tau_1}{2}\|v - \nabla u - b_1\|_2^2 \\ &\quad + \kappa\|w\|_1 - \langle p^T, w - u\rangle + \frac{\tau_2}{2}\|w - u - b_2\|_2^2. \end{aligned} \tag{25}$$

Let $X$ be the KKT condition as shown below.

$$\begin{cases} \lambda(f)A^T(Au^* - f) - p^* = 0 \\ \quad v^* - \nabla u^* = 0 \\ \quad w^* - u^* = 0 \\ \quad 0 \in \partial\|v^*\|_1 + y^* \\ \quad 0 \in \kappa\partial\|w^*\| + p^* \end{cases} \tag{26}$$

**Proof** . First, let $x^k = (u^k, v^k, w^k, b_1^k, b_2^k)$ be the iteration in Algorithm 1, $\tilde{x}^k = (u^k, v^k, w^k, \tau_1 b_1^k, \tau_2 b_2^k)$. The subproblem (15) about $u$ can be obtained by applying

SBIM to the problem (12), which in turn leads to the optimality condition (16). At this point, Equation (27) holds.

$$
\begin{cases}
\lambda(f)A^TA\left(u^{k+1}-u^k\right)+\tau_1\nabla^T\nabla\left(u^{k+1}-u^k\right)+\tau_2\left(u^{k+1}-u^k\right)\\
=\lambda(f)A^Tf+\tau_1\nabla^Tv^k-\tau_1\nabla^Tb_1^k+\tau_2w^k-\tau_2b_2^k-\lambda(f)A^TAu^k-\tau_1\nabla^T\nabla u^k-\tau_2u^k\\
v^{k+1}-v^k=shrink\left(\nabla u^k+b_1^k,\frac{1}{\tau_1}\right)-v^k\\
w^{k+1}-w^k=shrink\left(u^k+b_2^k,\frac{\kappa}{\tau_2}\right)-w^k\\
b_1^{k+1}-b_1^k=b_1^k+\nabla u^k-v^k\\
b_2^{k+1}-b_2^k=b_1^k+u^k-w^k
\end{cases}
\tag{27}
$$

It follows from Theorem 1 that $\lim_{k\to\infty}\left(x^k-x^{k+1}\right)=0$ and that the left-hand side of Equation (27) tends to 0 when $k\to\infty$, and, by extension, the right-hand side of Equation (27) also tends to 0. Therefore, when $k\to\infty$, all terms tend to be 0

$$
\begin{cases}
(\lambda(f)A^TAu^k+\tau_1\nabla^T\nabla u^k+\tau_2u^k-(\lambda(f)A^Tf\\
+\tau_1\nabla^Tv^k-\tau_1\nabla^Tb_1^k+\tau_2w^k-\tau_2b_2^k))\to 0\\
(shrink(\nabla u^k+b_1^k,\frac{1}{\tau_1})-v^k)\to 0\\
(shrink(u^k+b_2^k,\frac{\kappa}{\tau_2})-w^k)\to 0\\
(b_1^k+\nabla u^k-v^k)\to 0\\
(b_1^k+u^k-w^k)\to 0
\end{cases}
\tag{28}
$$

The following system of equations is easily obtained by analyzing the expressions (27) and (28).

$$
\begin{cases}
\lambda(f)A^TAu^*+\tau_1\nabla^T\nabla u^*+\tau_2u^*=\\
\lambda(f)A^Tf+\tau_1\nabla^Tv^*-\tau_1\nabla^Tb_1^*+\tau_2w^*-\tau_2b_2^*\\
v^*=shrink(\nabla u^*+b_1^*,\frac{1}{\tau_1})\\
w^*=shrink(u^*+b_2^*,\frac{\kappa}{\tau_2})
\end{cases}
\tag{29}
$$

In summary, it can be found that the KKT condition is satisfied for all accumulations concerning to $\widetilde{x}^k$. However, the KKT condition is the only necessary optimality condition for the non-convex optimization problem (11). Therefore, there is no guarantee that the optimal point of (24) is the point of accumulation. □

## 5. Numerical Experiments and Analysis

### 5.1. Image Dataset and Experimental Environment Setup

A Windows 10 system running with 8 CPUs of memory and MATLAB version R2018b is the experimental environment set up for this paper. In this paper, the denoising performance of the proposed model is evaluated using natural and artificial images with different resolutions. The images used are all grey-scale images and the natural images have multi-scale edges with rich texture structure, as shown in Figure 1.

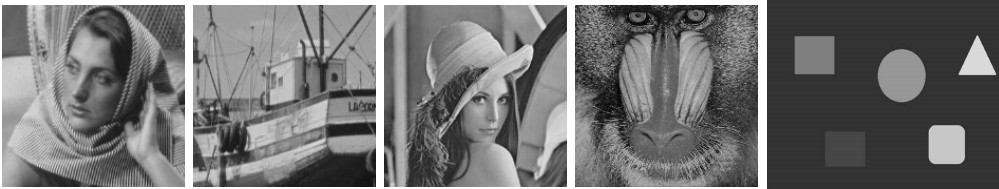

**Figure 1.** Experimental images.

### 5.2. Image Quality Assessment Indicators

Common metrics used to evaluate the quality of image recovery are peak signal-to-noise ratio (PSNR), structural self-similarity (SSIM), feature similarity, multi-scale structural similarity, and perceptual similarity. In this paper, SSIM and PSNR are used as evaluation metrics. the higher the PSNR value, the better the image recovery. The evaluation of SSIM

relies on the human visual system (HVS) and SSIM $\in [0, 1]$, the closer its value is to 1, the better the structure retention of the image. The relevant definitions are as follows.

$$\text{PSNR}(u^*, u) = 10 \log_{10}\left(\frac{255^2 MN}{\|u^* - u\|_2^2}\right) \tag{30}$$

$$\text{SSIM}(u^*, u) = \frac{2\mu_{u^*}\mu_u + C_1}{\mu_{u^*}^2 + \mu_u^2 + C_1} \cdot \frac{2\sigma_{u^*u} + C_2}{\sigma_{u^*}^2 + \sigma_u^2 + C_2} \tag{31}$$

Here, $u^*$ and $u$ denote the recovered image and the original image, respectively, $\mu$ denotes the mean, $\sigma$ denotes the covariance, $C_1$ and $C_2$ denote the constants, and $M$, $N$ denotes the maximum width and length of the image, respectively.

*5.3. Numerical Experiments*

Random Gaussian noise with mean 0 and variance $\sigma = 10, 20$ was added to all test images. The noisy images were processed for the first time using a mean filter and the result was used as the initial value for the algorithm. Next, the denoising performance was tested using the algorithm proposed in this paper. The proposed algorithm was compared with the PSNR and SSIM of related algorithms under the same PSNR and SSIM solution formulas to evaluate the algorithm's denoising performance, including LATV [34], TVAL3 [35], NGS [36], TVAL3 [37], and TVBH [11]. Experimenting with the algorithm of this paper on natural images, the PSNR and SSIM values of the algorithm can be obtained as shown below.

Adding random Gaussian noise with variance $\sigma = 10, 20$ to the natural image, the PSNR and SSIM values of this algorithm and the comparison test can be obtained as shown in Table 1, and the denoising effect is shown in Figures 2 and 3. Adding Gaussian noise with the variance of $\sigma = 20$ to the artificial image and using the TVBH model at the parameter ratio of 1/2, the denoising performance of the algorithm and the TVBH model is shown in Figure 4.

**Table 1.** PSNR and SSIM of different algorithms with different variances.

| Delta | Methods | Evaluating Criterion (PSNR/SSIM) | | | |
|---|---|---|---|---|---|
| | | **Lena** | **Barbara** | **Boats** | **Baboon** |
| Delta = 10 | LATV | 32.2371/0.8834 | 29.8621/0.8824 | 31.1085/0.8931 | 27.5627/0.8906 |
| | T-ASTV | 32.4306/0.8901 | 29.5901/0.8913 | 31.1069/0.8947 | 27.2326/**0.8952** |
| | NGS | 32.4195/0.8983 | 30.2378/**0.8976** | 31.0814/0.8943 | 28.0918/0.8917 |
| | TVAL3 | 32.3914/0.8843 | 30.1947/0.8862 | 31.1716/0.9013 | 28.0125/0.8922 |
| | ours | **32.4321/0.8987** | **30.4974**/0.8932 | **31.8834/0.9029** | **28.3017**/0.8923 |
| Delta = 20 | LATV | 28.1323/0.8906 | 25.0115/**0.9029** | 27.3971/0.9012 | 25.1216/0.9023 |
| | T-ASTV | 28.4741/0.8878 | 25.5346/0.8989 | 27.4461/0.9063 | 24.9958/0.9046 |
| | NGS | 28.4552/0.8924 | 25.8304/0.8968 | 27.3014/0.9103 | 24.8649/0.9014 |
| | TVAL3 | 28.4545/0.8929 | 25.8467/0.8994 | 27.40130.9046 | 25.1246/**0.9127** |
| | ours | **28.4938/0.9050** | **26.8427**/0.9009 | **27.8708/0.9068** | **25.2037**/0.9091 |

Note: The bolded font is the optimal value.

In Table 1, when $\sigma = 10$, the PSNR and SSIM values for each algorithm are higher for Lena, Barbara, Boats, and Baboon, but the proposed algorithm has higher values for the evaluation metrics than the other algorithms in most cases. When $\sigma = 20$, the PSNR and SSIM values of each algorithm decreased, but the proposed algorithm's PSNR and SSIM were still slightly higher than those of the other models. Therefore, the denoising performance of the proposed algorithm has been improved.

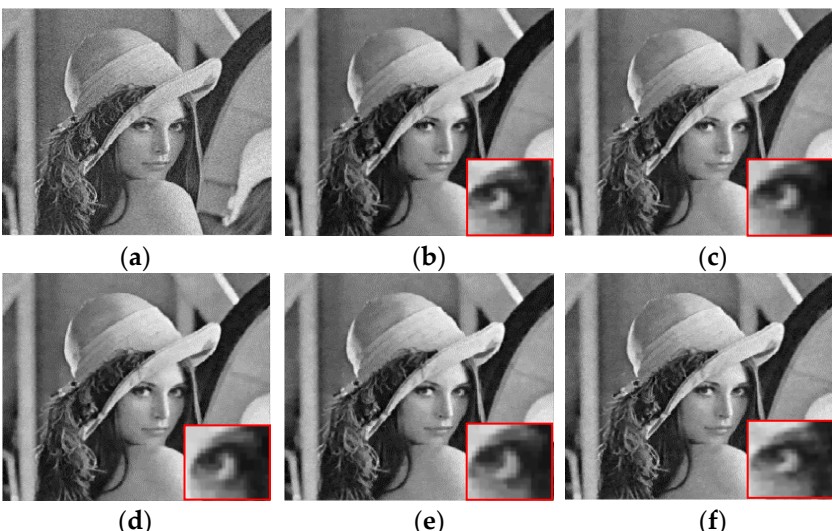

**Figure 2.** The denoising effect of each algorithm on Lena for $\sigma = 10$. Where (**a**) shows the noisy image with $\sigma = 10$, (**b**–**e**) show the denoising effect of LATV, T-ASTV, T-ASTV, and TVAL3 algorithms, respectively, and (**f**) shows the denoising effect of the algorithm in this paper.

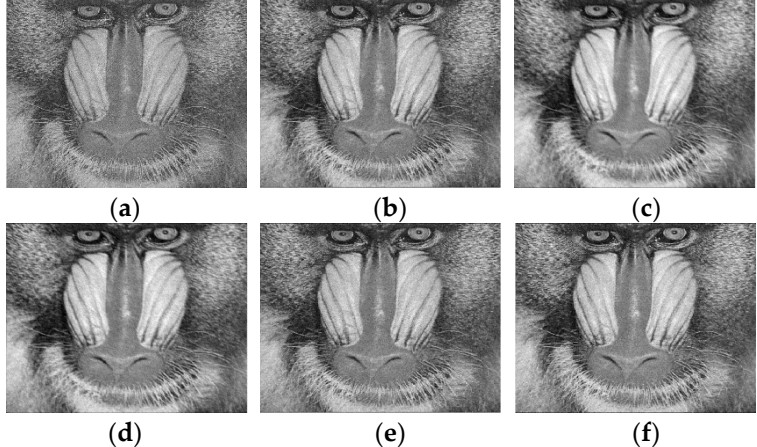

**Figure 3.** The denoising effect of each algorithm on Lena for $\sigma = 20$. Where (**a**) shows the noisy image with $\sigma = 20$, (**b**–**e**) show the denoising effect of LATV, T-ASTV, T-ASTV, and TVAL3 algorithms, respectively, and (**f**) shows the denoising effect of the algorithm in this paper.

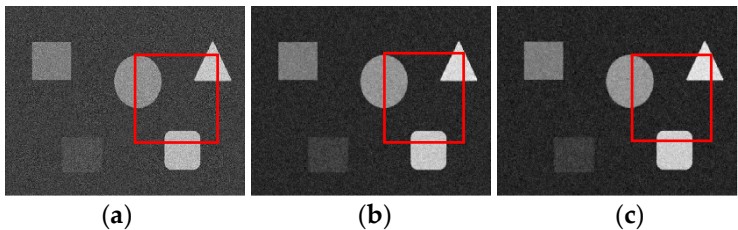

**Figure 4.** The denoising effect of artificial images on our algorithm and TVBH algorithm. Where (**a**) shows the noisy image with $\sigma = 20$, (**b**), show the denoising effect of TVBH model, and (**c**) shows the denoising effect of the algorithm in this paper.

Figure 2a shows the noisy image with $\sigma = 10$, Figure 2b,e show the denoising effect of LATV, T-ASTV, T-ASTV, and TVAL3 algorithms, respectively, and Figure 2f shows the denoising effect of the algorithm in this paper. In the red boxed line is the enlarged right eye part. Overall, the denoising effect of each model is acceptable, but the clarity of Figure 2e

is slightly higher than that of Figure 2b–d. Figure 2f also has higher clarity, looking at the middle of the hat, where the texture is more obvious looking at the red boxed part, and Figure 2b,e show a more obvious step effect, while Figure 2f also shows a step effect, but to a lesser extent than Figure 2b,e. Thus, the proposed algorithm improves the step artifact effect of the model.

Figure 3 presents the denoising effect of each model on the baboon for variance $\sigma = 20$. Figure 3a shows the noisy image with variance, Figure 3b–e show the denoising effect of the LATV, T-ASTV, T-ASTV, and TVAL3 algorithms, respectively, and Figure 3f shows the denoising effect of the algorithms in this paper. It can be found that Figure 3b,e remove some noise, but there is still a lot of noise remaining in the image. In contrast, Figure 3c,d remove more noise, but there is blurring. A closer look shows that Figure 3f removes more noise, while the image is not blurred and shows clearer texture detail.

In Figure 4, Figure 4a shows the noisy image with $\sigma = 20$, Figure 4b shows the denoised image of the TVBH model, and Figure 4c shows the denoised image of the proposed algorithm. Looking at the red boxed parts, we find that the whiteness of the triangles, circles, and squares in Figure 4c is more obvious than in Figure 4a,b, indicating that the proposed algorithm removes more Gaussian noise. Therefore, the proposed model noise filtering is better.

## 6. Conclusions

In this paper, we propose an image denoising algorithm for multi-scale parameter estimation, taking advantage of the fact that the TV regularization term can remove noise, the preservation of edges, and the $L_1$ norm regularization term can promote the sparsity of the model solution and enhance the model to suppress step artifact effects. In the algorithm, we only need to estimate the value of one parameter $\kappa$, which effectively reduces the complexity of the estimated parameters. Furthermore, based on the values of PSNR and SSIM of the proposed algorithm and the comparison algorithm, it can be determined that the proposed algorithm has better denoising performance and the robustness of the model to noise is enhanced. Moreover, based on the denoising effect plots, it can be found that the step artifact effect of the image is suppressed and the noise filtering effect is enhanced. Overall, the proposed algorithm has a better denoising performance and outperforms the comparison model.

In the experiments, it can be found that although the model proposed in this paper achieves good results, there are still some problems, such as the presence of a small amount of noise in the image or the incomplete preservation of image details. In future research work, on the one hand, some new concepts related to fuzzy fractional calculus [38], and Hermite–Hadamard Inequalities [39–41] can be used instead of the concepts associated with existing models to improve existing algorithms and thus improve the image denoising performance of the models. On the other hand, the edge detection function can be applied to detect the edges of the image, thus better preserving the texture details of the image [42].

**Author Contributions:** Conceptualization, M.L.; methodology, M.L.; soft, M.L. and S.B.; validation, M.L.; formal analysis, M.L.; investigation, M.L.; resources, M.L.; data curation, M.L., S.B., G.C. and X.Z.; writing—original draft preparation, M.L.; writing—review and editing, M.L.; visualization, M.L., supervision, S.B., G.C. and X.Z. project administration, M.L.; funding acquisition, G.C. All authors have read and agreed to the published version of the manuscript.

**Funding:** National Natural Science Foundation of China (11461037) and High Quality Postgraduate Courses of Yunnan Province (109920210027).

**Data Availability Statement:** Experimental data for tis study can be obtained by looking up or contacting the authors on GitHub.

**Acknowledgments:** We thank the editor and anonymous reviewers for their valuable comments and suggestions on our manuscript. National Natural Science Foundation of China (11461037) and High Quality Postgraduate Courses of Yunnan Province (109920210027) are gratefully acknowledged.

**Conflicts of Interest:** The authors declare that they have no known competing financial interests or personal relationships that could have appeared to influence the work reported in this paper.

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
