# Peer review of "Improved TV Image Denoising over Inverse Gradient"

_symmetry, doi:10.3390/sym15030678_

Round 1

Reviewer 1 Report

Improved TV image denoising based on inverse gradient

Minmin Li, Guangcheng Cai, Shaojiu Bi, Xi Zhang

Authors proposed an image denoising algorithm for multi-scale parameter estimation, taking advantage of the fact that the TV regularization term can remove noise, the preservation of edges and the L1 norm regularization term can promote the sparsity of the model solution and enhance the model to suppress step artifact effects. In the algorithm, they only need to estimate the value of one parameter k , which effectively reduces the complexity of the estimated parameters. Furthermore, based on the values of PSNR and SSIM of the proposed algorithm and the comparison algorithm, it can be determined that the proposed algorithm has better denoising performance and the robustness of the model to noise is enhanced. Also, based on the denoising effect plots, it is found that the step artifact effect of the image is suppressed, and the noise filtering effect is enhanced. Overall, the proposed algorithm has a better denoising performance and outperforms the comparison model.

The results are mathematically and computationally are correct to the best of my knowledge.

Author Response

We are very grateful for the professional suggestions and comments provided by the reviewer.

Author Response

We feel great thanks for your professional review work on our article. As you are concerned, there are several problems that need to be addressed. According to your nice suggestions, we have made extensive corrections to our previous draft, the detailed corrections are listed below.

  1. According to the reviewer's suggestion, we have substantially revised the introduction of this manuscript. The concept of "symmetry" and its use in image processing has been added as the first paragraph in the introduction. In the Second and third paragraphs, we describe the references and their results in detail.
  2. As suggested by the reviewers, additional references have been added where appropriate to support the conclusions of this paper and to provide a detailed description of the literature.
  3. Thanks for your careful checks. We have revised the language issues you raised and have also re-reviewed this article.

We would like to thank the referees for their valuable comments and suggestions. We hope that the revision is acceptable, and I look forward to hearing from you soon.

Author Response

We feel great thanks for your professional review work on our article. As you are concerned, there are several problems that need to be addressed. According to your nice suggestions, we have made extensive corrections to our previous draft, the detailed corrections are listed below.

Point (1). I will suggest the title “On “Improved TV image denoising over inverse gradient” instead of “Improved TV image denoising based on inverse gradient”.

Response 1: Based on the valuable comments you have provided, and in consultation with the other authors, we have unanimously decided to revise the title of our paper to: Improved TV image denoising based on inverse gradient.

Point (2). To improve the impact and readership of your manuscript, the authors need to clearly articulate in the Abstract and the Introduction sections the uniqueness or novelty of this article, and why or how it is different from other similar articles.

Response 2: According to the reviewer's suggestion, we have substantially revised the Abstract and introduction of this manuscript. A brief explanation of the uniqueness of this article and how it different from other similar articles.

Point (3). The Conclusion section is too short. Please expand it by discussing the future directions of your research, especially how it may contribute to your ongoing research about "symmetry".

Response 3: We think this is an excellent suggestion. We have expanded the concluding section of the paper accordingly, and apply about "symmetry" to future research.

Point (4). Please substantially expand your review work, and cite more of the journal papers published by MDPI.

Response 4: We have carefully reviewed the paper and added relevant literature to the introduction and conclusion sections of the revised manuscript.

Point (5). Some of the references cited are not yet properly formatted. For example, For the references, instead of formatting "by-hand", please kindly consider using the free Zotero software (https://www.zotero.org/), and select "Multidisciplinary Digital Publishing Institute" as the citation format, since there are currently 20 citations in your manuscript, and there may probably be more once you have revised the manuscript. I will suggest following article need to be cite. Some new concepts related to fuzzy fractional calculus for up and down convex fuzzy-number valued functions and inequalities; New Hermite–Hadamard Inequalities for Convex Fuzzy-Number-Valued Mappings via Fuzzy Riemann Integrals; Hermite-Hadamard inequalities for generalized convex functions in interval-valued calculus; Generalized Preinvex Interval-Valued Functions and Related Hermite–Hadamard Type Inequalities.

Response 5: Manually entering references is indeed a rather unwieldy method and is prone to citation formatting errors. We have therefore re-entered the references using Zotero software and applied the relevant literature to the thesis. However, it is necessary to explain that some of the references cited in this manuscript only propose methods in theory and are not applied to practical problems. Therefore, the above references may lack profound results obtained after solving practical problems, and we hope to get the reviewer's understanding and consideration.

We would like to thank the referees for their valuable comments and suggestions. We hope that the revision is acceptable, and I look forward to hearing from you soon.

Round 2

Reviewer 2 Report

Accept

Reviewer 3 Report

Thank for your nice article